# Personal Accomplishment and Hardiness in Reducing Emergency Stress and Burnout among COVID-19 Emergency Workers

**Monia Vagni**, **Valeria Giostra, Tiziana Maiorano**, **Giuliano Santaniello and Daniela Pajardi** *

Department of Humanities, University of Urbino, 61029 Urbino, Italy; monia.vagni@uniurb.it (M.V.);
valeria.giostra@uniurb.it (V.G.); tiz.maiorano@gmail.com (T.M.); g.santaniello@campus.uniurb.it (G.S.)

*  Correspondence: daniela.pajardi@uniurb.it; Tel.: +39-0722-305814

**Abstract:** During the severe phase of the pandemic, COVID-19 emergency workers were engaged in long and numerous shifts of duty, resulting in exposure to various stress factors. A high stress level is associated with risk of burnout. Resilience and personal accomplishment can effectively help mitigate and reduce emergency stress levels and emotional exhaustion. The main aim of this study was to analyze the relationship of emergency stress and hardiness with burnout among emergency workers. The participants included 494 emergency volunteers from the Red Cross Committee in Veneto, Italy, engaged in various health, emergency, and social activities aimed at COVID-19 patients and people at risk of contracting the virus. Questionnaires used to measure emergency stress, hardiness and burnout were administered on an online platform. We analyzed the influence of age, sex, weekly hours of service, stress risk factors, and use of personal protective equipment. To verify the predictive effects of risk and protective factors on burnout, correlational and multivariate analyses, and regressions were conducted. Hardiness showed an effect in reducing emergency stress levels, emotional exhaustion, and depersonalization and simultaneously increased personal accomplishment.

**Keywords:** emergency stress; hardiness; burnout; emergency workers; resilience; COVID-19; mental health

---

## 1. Introduction

During the first wave of the COVID-19 pandemic, Italian volunteer emergency workers intervened in several situations to help with the urgent transport of the sick and with creating the necessary aids to provide medicine, assistance, and basic necessities. Some volunteer emergency workers offered their help for many hours a week, working many long shifts, reducing their hours of rest and exposing themselves to the risk of virus contagion. The Red Cross volunteers are highly specialized rescuers, but the COVID-19 emergency, being new and unexpected, necessitated the rapid construction and implementation of new and specific intervention protocols. Since there was no previous experience or knowledge of the virus, how the interventions and the situation were perceived was linked to the subjective and personal cognitive and emotional categories.

Because of the wide spread of the virus, the protection of healthcare personnel from the risk of burnout, which has exceeded the estimates previously recorded [1], has become a primary concern. The main purpose of this study was to detect the risk of emergency workers developing burnout and the risk and protection factors that contribute to its determination.

### 1.1. Stress and Burnout in Emergency Workers

Emergency workers are exposed to emotionally challenging [2] and potentially traumatic situations that may develop into high levels of stress, with mental and physical changes [3], such as

irritability, fatigue, emotional exhaustion, sleep disorders, problems with interpersonal relationships [4], and concerns for individual safety [5]. The literature highlights how perceiving high levels of stress in the absence of support may lead to mental disorders such as vicarious traumatization in emergency workers [6]. A few studies have considered the differences in stress reactions between professional and volunteer emergency workers [7], reporting that there are no significant differences between the two groups if the volunteers follow exhaustive and adequate training and preparation courses.

Emergency workers involved in the COVID-19 pandemic have been exposed to high levels of stressful or traumatic events, which can lead to negative mental health outcomes, including stress-related symptoms and symptoms of depression, anxiety, insomnia, and vicarious traumatization [8,9]. Some studies highlighted that the female sex and a lack of personal protective equipment (PPE) are important risk factors for the development of high levels of perceived stress. A study conducted in Spain found an increased risk of developing high stress in ambulance personnel who had treated COVID-19 patients [10].

To cope with stressful situations, emergency workers may use counterfactual thinking, which can be defined as mentally changing the perceived antecedents of an outcome and mentally playing out the consequences once the facts are known. Whereas downward counterfactuals tend to follow satisfying outcomes, leading to a relatively positive affect [11], upward counterfactuals tend to lead to a relatively negative affect. Counterfactual thinking can intervene in the decision-making processes of everyday life and is practiced both in common sense and by experts [12,13] but is typically activated by negative experiences or, more generally, by factors related to goal blockage (e.g., perception of a problem, disconfirmed expectancy). Therefore, in emergency situations such as the COVID-19 pandemic, which are characterized by unexpected events, lack of clear intervention protocols, and lack of resources, emergency workers are more likely to use upward counterfactual thinking. As several studies have highlighted that counterfactual thinking can be associated with mental states of regret, rumination, and amplified negative emotions [12,14–17], it can be a risk factor for developing higher levels of stress and psychopathological symptoms [12].

High levels of perceived stress within the work setting can result in burnout. Burnout is defined as a mental condition arising as the body's response to the failure of coping mechanisms that people generally use to manage stress at work [18–20]. The buildup of stress exhausts people, so whenever their energy resources are low, they attempt to overcome the pressure during their work with others [21,22]. Employees lose the interest and positive feelings they had for the people they are assisting and develop a negative self-image [23]. Freudenberger [24] was the first to outline the concept of professional burnout, but the foremost and widely-accepted definition of burnout was developed by Maslach [23]. Maslach defined it as a psychological syndrome of emotional exhaustion, depersonalization, and reduced personal accomplishment that develops in those who have an expert relationship with other persons. The workers who develop burnout are characterized by chronic exhaustion, impotency, and cynicism [25]. This syndrome produces a robust and negative impact on workers' physical and psychological health and on their organization and work, affecting both health and emergency operators [26,27].

The literature shows that caring for extremely large numbers of patients, the deaths of patients, disturbances in sleep patterns, and long working hours are major stressors. In addition, handling patients presenting with confounding and challenging diagnoses and treatments, problems with professional relationships, inadequate supply of resources, making mistakes with patients' companions, unsafe working environments, and difficult and critical judgments play important roles in the development of burnout [28]. As mentioned above, many of these risk factors have been present during the COVID-19 epidemic. Furthermore, burnout usually occurs following long-term exposure to organizational risk factors, and the literature shows that critical emergencies, like pandemics, can easily trigger emotional exhaustion [29].

Although emergency workers are being exposed to high levels of stress and risk factors for developing burnout, at present, no studies have investigated the levels of burnout in emergency

workers involved in the treatment and management of COVID-19 patients. Studies on burnout related to COVID-19 have been mainly conducted on medical staff, reporting high levels of perceived stress, excessive work shifts, and the lack of PPE as risk factors. A study on 2707 healthcare professionals (HCPs) from 60 countries showed that burnout is significantly associated with high stress levels, high workloads, high time pressure, limited access to PPE, and exposure to COVID-19 patients [30]. Exposure to COVID-19 patients and being female were highlighted as risk factors by Kannampallil et al. [31] in a sample of physician trainees. Another study found a significant relationship between burnout, anxiety, and stress disorders among doctors and nurses who treated COVID-19 patients [32]. A study on Italian healthcare professionals [33] reported relevant work-related psychological pressure, somatic symptoms, and emotional burnout; a large percentage of healthcare professionals reported high scores in at least one of the Maslach Burnout Inventory (MBI) domains. In particular, more than one out of three reported a high emotional exhaustion score, with women reporting higher levels than their male counterparts, and one out of four reported high levels of depersonalization, while only around 15% reported low levels of personal gratification. Giusti et al. [34] found that work hours, psychological comorbidities, female sex, working in the hospital, fear of infection, and being in contact with COVID-19 patients are predictors of both emotional exhaustion and depersonalization of burnout.

### 1.2. Protective Factors against Stress and Burnout: Resilience and Hardiness

According to several studies on emergency situations, it is important to strengthen resilient abilities that may increase the process of adaptation to negative life events [35]. Resilience can allow flexible adaptation to a new situation [36] and can contribute to positive functioning, preventing negative emotions, thoughts, and behaviors [37]. Studies about resilience during the COVID-19 pandemic have shown that high resilience prevents and reduces negative affect, depression, and anxiety, and may increase positive affect, life satisfaction, and subjective well-being in both healthcare workers [38] and the general population [39]. Resilience can be a effective mediator between stress and burnout, acting in emergency workers, in particular, as a protective factor that prevents first responders from developing symptoms of stress, burnout, and vicarious traumatization [40–44].

Resilience is composed of multiple factors (e.g., hardiness, self-enhancement, repressive coping, positive emotion, and laughter) and is derived from various pathways [45]. Hardiness, which is also known as dispositional resilience, is a style of functioning that distinguishes people who remain healthy in response to stress from those who develop stress-related health problems [46,47]. Bartone [48] (p. 137) describes hardy individuals as those who have "a high sense of life and work commitment, a greater feeling of control, and are more open to change and challenges in life" than others. These three characteristics were named by Kobasa [47] as: commitment, control, and challenge.

The effects of stress and burnout seem to be reduced by hardiness, which can be defined a protective factor [49,50]. For example, a study on ambulance personnel [49] showed that workers with high levels of hardiness, measured by the hardiness scale (HS) [51], are less likely to display general psychopathology, post-traumatic symptoms, and burnout. In particular, the MBI subscales of emotional exhaustion and positive accomplishment were found to be significantly associated with all three HS sub-scales, whereas depersonalization showed a negative correlation only with the commitment and control subscales.

Regarding the COVID-19 pandemic, some studies among healthcare workers [52–54] found resilience to be a protective factor against adverse mental health outcomes. Among Italian emergency workers, hardiness emerged as a mediator, reducing the predicted effect of stress on vicarious trauma [9]. To date, however, the effects of hardiness on the burnout of emergency workers have not been investigated.

## 2. Materials and Methods

The main objective of this study was to identify the predictive effect of some risk factors and emergency stress on burnout perceived by COVID-19 volunteer emergency workers.

Emergency situations may lead to emotional, cognitive, physical, and social relational reactions [55], but can also result in ineffective decision-making, and, in the specific case of the COVID-19 pandemic, the fears of contracting the virus and of infecting their own families were specifically considered [55–57].

Our main hypotheses were:

**Hypothesis 1 (H1).** *Emergency stress is positively correlated with burnout.*

**Hypothesis 2 (H2).** *Emergency stress is negatively correlated with hardiness and personal accomplishment.*

**Hypothesis 3 (H3).** *Risk factors have a predictive effect on emergency stress.*

**Hypothesis 4 (H4).** *Hardiness has a protective effect on emergency stress.*

**Hypothesis 5 (H5).** *Hardiness reduces the predictive effect of emergency stress on burnout.*

The regression models used to verify Hypotheses 2 and 3 were checked by the other three variables: sex, age, and COVID-19 patients.

### 2.1. Participants

The sample comprised 494 Red Cross volunteers from the Veneto Regional Committee, Italy. Of the participants, 280 (56.7%) were women (mean age = 44.40, SD 12.92; range 18–75) and the remaining 214 were men (mean age = 47.25, SD 13.52; range 18–75). On average, the male participants were older than the female participants ($t = 2.38$; $p < 0.05$). The distribution of the sex variable within the sample was significant ($\chi^2 = 8.82$; $p < 0.01$). Volunteer emergency workers worked, on average, 13.05 h per week (SD 12.26, range 1–84). Length of service by week showed no differences between women and men.

Participants were assigned to three different intervention projects based on their training and specialization: "Health" ($n = 186$, 37.7%), which involved emergency room interventions, transport of the sick, etc.; "Social" ($n = 151$, 30.6%), where they provided social support and inclusion actions; and "Emergency" ($n = 157$, 31.7%), which involved the management of emergency units and COVID-points open during the emergency.

The distribution of the project variables within the sample was not significant. Among the interventions, 39.7% of the participants treated patients with a clear diagnosis of COVID-19 ("COVID-19 patients" variable).

### 2.2. Procedure

An online transactional survey was used in the second phase of the pandemic. The questionnaire included: informed consent, baseline sociodemographic information, and the instruments used to detect emergency stress, hardiness, and burnout. Information about some risk factors considered relevant in this study was also collected. The institutional ethics committee approved all of the procedures used.

### 2.3. Materials

In this study, we administered the following instruments:

(1)    Emergency Stress Questionnaire (ESQ) [8,9,58]: This self-reported questionnaire, already published and validated, measured the level of stress in medical staff and emergency workers during phases 1 and 2 of the pandemic [8,9,58]. The ESQ consists of 33 items assessed on a five-point Likert scale, with scores ranging from 0 (not at all) to 4 (very much). Factor analysis was conducted, and confirmatory factor analysis (CFA) confirmed the presence of the six scales or factors. The internal consistency was good, and the following were found for each scale: Cronbach's

alpha was 0.71, 0.82, 0.80, 0.86, 0.72, 0.80, and 0.93 for organizational–relational stress, physical stress, inefficacy-decisional stress, emotional stress, cognitive stress, COVID-19 stress, and total ESQ, respectively.

(2)　Dispositional Resilience Scale-15, Italian version (DRS-15) [59,60]: This self-reported questionnaire consists of 15 items scored on a four-point scale ranging from 0 (not at all true) to 3 (completely true), and measures the hardiness using three scales: commitment, control, and challenge. The overall score ranges from 0 to 45, with higher scores indicating a higher level of hardiness. In addition to the total score, the DRS yields scores for the three subscales. Italian standardization showed low alpha coefficient values except for total DRS and challenge (0.72 and 0.70, respectively). We considered it appropriate to use only the total DRS scores.

(3)　Maslach Burnout Inventory–Human Services Survey, Italian version (MBI–HSS) [61–63]: This is a self-reported instrument that measures respondents' perceived experience of burnout in relation to the recipients of their service, care, or treatment. The MBI–HSS consists of 20 items on a frequency scale ranging from 0 (never) to 6 (every day). Three subscales can be identified: emotional exhaustion, depersonalization, and personal accomplishment. Emotional exhaustion refers to the feeling that emotional resources are drained and a scarcity of energy (nine items, e.g., "I feel emotionally drained from my work"). Depersonalization refers to the perception of behaving toward the recipients of the service in a detached, cynical, and emotionally callous manner (five items, e.g., "I feel I treat some recipients as if they are impersonal objects"). Reduced personal accomplishment refers to the subject's sense of achievement at work (eight items, e.g., "I have accomplished many worthwhile things in this job"). High scores for the first two scales and low scores for the last one are associated with burnout.

(4)　An original questionnaire/checklist on stressful factors: A checklist of seven items was built in dichotomous form (yes/no). These items were considered in the literature as factors that can increase stress reactions because they increase the feeling of loss of control and reduce the sense of self-efficacy. Rescuers, following their interventions, often participate in debriefings to evaluate the effectiveness of the intervention, also activating a counterfactual reasoning that can, in some cases, increase the sense of frustration associated with greater reactions of emotional, cognitive, and decision-making stress [11,12,55,64,65]. The checklist included yes/no questions to detect stress factors identified by the literature, such as:

- 　Having suggested solutions that have not been considered ("Suggestions"),
- 　Having experienced unexpected and unpredictable events ("Unexpected-events"),
- 　Having received the necessary instructions to intervene ("Instructions"),
- 　Having PPE,
- 　Having made a decision that proved ineffective ("Ineffective-decision"),
- 　Believing in hindsight that it would have been appropriate to intervene in a different way ("Different-behavior"), and
- 　Having received unclear information ("Unclear information") was not considered due to missing values.

The average scores and standard deviations of the ESQ, DRS-15, and MBI–HSS are shown in Appendix A, Table A1, and the distribution of the risk factors is shown in Table A2.

## 2.4. Statistical Analysis

Pearson's correlation was conducted to detect the presence of associations among all scales examined in this study. A multivariate analysis was performed to verify the effects of service/project by participants groups, sex, age, weekly working hours, and COVID-19 patients on the levels of stress, hardiness, and burnout.

We used hierarchical linear regression to verify both the predictive effect of the risk factors on the levels of the emergency stress (step 1) and the protective effect of hardiness (step 2). The model was controlled for age, sex, and COVID-19 patients (step 1).

Subsequently, a hierarchical linear regression model was performed to verify the predictivity power of hardiness and emergency stress scales on burnout. The model was controlled for age, sex, and COVID-19 patients.

## 3. Results

### 3.1. Preliminary Analysis

A Pearson's correlation was performed between the variables of interest in this study: ESQ scales, total DRS, and MBI–HSS to verify Hypothesis 1 (Table 1). The results confirmed a significative positive correlation between emergency stress and burnout (H1) and significative negative correlations of emergency stress with hardiness and personal accomplishment (H2).

**Table 1.** Pearson's correlations between ESQ, DRS-15, and MBI–HSS ($n$ = 494).

|  | DRS | MBI–HSS | | |
|---|---|---|---|---|
|  | Total DRS | Emotional Exhaustion | Depersonalization | Personal Accomplishment |
| **ESQ** |  |  |  |  |
| Organizational–Relational Stress | −0.215 *** | 0.520 *** | 0.308 *** | −0.196 *** |
| Physical Stress | −0.311 *** | 0.616 *** | 0.265 *** | −0.180 *** |
| Inefficacy-Decisional Stress | −0.027 | 0.324 *** | 0.172 *** | 0.012 |
| Emotional Stress | −0.333 *** | 0.565 *** | 0.290 *** | −0.320 *** |
| Cognitive Stress | −0.278 *** | 0.543 *** | 0.346 *** | −0.288 *** |
| Covid-19 Stress | −0.135 ** | 0.419 *** | 0.200 *** | −0.139 ** |
| **DRS-15 (total)** | 1 | −0.273 *** | −0.205 *** | 0.343 *** |
| **MBI–HSS** |  |  |  |  |
| Emotional Exhaustion | −0.276 *** | 1 | 0.477 *** | −0.261 *** |
| Depersonalization | −0.205 *** | 0.477 *** | 1 | −0.204 *** |
| Personal Accomplishment | 0.343 *** | −0.261 *** | −0.204 *** | 1 |

Note: ** $p < 0.01$, *** $p < 0.001$; ESQ, Emergency Stress Questionnaire; DRS-15, Dispositional Resilience Scale; MBI–HSS, Maslach Burnout Inventory–Human Services Survey.

### 3.2. Results

Multivariate analysis (MANOVA) was performed assuming levels of stress, hardiness, and burnout as dependent variables, and COVID-19 patients and project group (Health, Social, or Emergency) as fixed factors. Age, sex, and weekly working hours were covariates.

The model showed significant within-subject effects related to the COVID-19 patients group (Pillai's value = 0.072, F = 3.466, *df* (10, 444), $p < 0.001$, $\eta^2$ = 0.072), age (Pillai's value = 0.056, F = 2.651, *df* (10, 444), $p < 0.01$, $\eta^2$ = 0.056), and sex (Pillai's value = 0.082, F = 3.951, *df* (10, 444), $p < 0.001$, $\eta^2$ = 0.082). Weekly working hours and projects were not significant, and the project × COVID-19 patients interaction did not assume significance.

Within the subjects, the results showed an effect of sex, with women reporting higher levels of physical (F = 19.402, $p < 0.001$, $\eta^2$ = 0.041, estimate of parameters $t$ = 4.404, $p < 0.001$) and emotional stress (F = 3.688, $p < 0.05$, $\eta^2$ = 0.011, estimate of parameters $t$ = 1.920, $p < 0.05$) than men.

Older volunteer emergency workers showed higher levels of organizational−relational (F = 4.840, $p < 0.05$, $\eta^2$ =0.011, $t$ = −2.200, $p < 0.05$), physical (F = 7.865, $p < 0.01$, $\eta^2$ 0.017, $t$ = −2.804, $p < 0.01$), and emotional stress (F = 8.674, $p < 0.01$, $\eta^2$ 0.019, $t$ = −2.945, $p < 0.01$). They simultaneously seemed to

report lower personal accomplishment (F = 7.240, $p < 0.01$, $\eta^2 = 0.016$, $t = 2.691$, $p < 0.01$) and hardiness scores (F = 11.867, $p < 0.01$, $\eta^2 = 0.026$, $t = 3.445$, $p < 0.01$).

Project group showed an effect within subjects at the levels of COVID-19 stress (F = 4.129, $p < 0.05$, $\eta^2 = 0.018$) and personal accomplishment (F = 3.467, $p < 0.05$, $\eta^2 = 0.015$). Emergency workers who treated COVID-19 patients reported higher inefficacy-decision stress (F = 18.038, $p < 0.001$, $\eta^2 = 0.038$, $t = 2.991$, $p < 0.001$) but also higher personal accomplishment (F = 13.148, $p < 0.001$, $\eta^2 = 0.028$, $t = 2.261$, $p < 0.05$).

The distribution of risk factors in the emergency workers who treated COVID-19 patients was verified: 124 of them reported unexpected-events ($\chi^2 = 7.180$; $p < 0.01$) and 153 did not have ineffective-decision ($\chi^2 = 4.602$; $p < 0.05$). For the other variables, we found no differences in their distribution within the COVID-19 patients vs. the non-COVID-19 patients group. Appendix A, Table A2 shows the distributions for each risk factor.

Hypotheses 3 and 4 were verified by regression analysis, considering the effects recorded in the MANOVA analysis, which showed the significative effects of the COVID-19 group, age, and sex variables on emergency stress. Hierarchical linear regression models were generated assuming each emergency stress scale as a dependent variable, and risk factors and hardiness were assumed as predictors in steps 1 and step 2, respectively. The models were checked by age, sex, and COVID-19 patients group (step 1). Unexpected-events showed a significative effect on all the emergency stress scales. The lack of instructions showed a predictive effect for all the emergency stress scales, except for the COVID-19 stress scale. Ineffective-decision was found to be a predictor of the organizational–relational, inefficacy, emotional, and cognitive stresses. The lack of PPE showed predictivity on organizational–relational and cognitive stress; and having suggested solutions that were not considered (suggestions) was found to be a predictor only of organizational–relational stress. The results shown in Table 2 confirmed Hypothesis 3. In step 2, hardiness showed a negative predictive effect for all the emergency stress scales except for inefficacy-decisional stress. The negative predictive effect of hardiness on emergency stress scales supported Hypothesis 4.

**Table 2.** Hierarchical linear regression models on ESQ scales ($n = 494$).

| | Organizational–Relational Stress | | Physical Stress | | Inefficacy-Decisional Stress | | Emotional Stress | | Cognitive Stress | | COVID-19 Stress | |
|---|---|---|---|---|---|---|---|---|---|---|---|---|
| | **B** | **Exp(B)** | **B** | **Exp(B)** | **B** | **Exp(B)** | **B** | **Exp(B)** | **B** | **Exp(B)** | **B** | **Exp(B)** |
| **Model 1** | | | | | | | | | | | | |
| Age | −0.53 | −0.151 *** | −0.62 | −0.169 *** | −0.28 | −0.097 * | −0.056 | −0.166 *** | −0.034 | −0.168 *** | −0.026 | −0.079 |
| Sex [1] | 0.518 | 0.056 | 1.975 | 0.202 *** | −0.209 | −0.027 | 0.903 | 0.101 * | 0.452 | 0.084 * | 0.250 | 0.029 |
| COVID-19 patients [2] | 0.061 | 0.007 | 0.197 | 0.020 | −1.103 | −0.143 ** | −0.108 | −0.012 | 0.126 | 0.023 | −0.560 | −0.064 |
| Suggestions [3] | −1.111 | −0.120 ** | −0.561 | −0.058 | −0.105 | −0.014 | 0.321 | 0.036 | −0.432 | −0.080 | −0.095 | −0.011 |
| Unexpected-events [4] | −1.601 | −0.173 *** | −1.005 | −0.103 * | −1.820 | −0.239 *** | −0.770 | −0.086 * | −0.640 | −0.119 ** | −1.114 | −0.130 ** |
| Instructions [5] | 1.642 | 0.161 ** | 1.070 | 0.099 * | 1.307 | 0.155 ** | 1.722 | 0.175 ** | 0.569 | 0.096 * | 0.754 | 0.079 |
| PPE [6] | 2.038 | 0.175 *** | 0.879 | 0.072 | −0.155 | −0.016 | 0.521 | 0.046 | 0.695 | 0.103 * | 0.109 | 0.010 |
| Ineffective-decision [7] | −1.164 | −0.095 * | −0.711 | −0.55 | −1.446 | −0.143 ** | −1.754 | −0.148 ** | −1.347 | −0.189 *** | −0.765 | −0.067 |
| Different-behavior [8] | −0.940 | −0.100 * | −0.154 | −0.016 | −0.509 | −0.066 | −0.355 | −0.039 | −1.104 | −0.202 *** | −0.198 | −0.023 |
| | $R^2 = 0.264$ | | $R^2 = 0.131$ | | $R^2 = 0.208$ | | $R^2 = 0.131$ | | $R^2 = 0.252$ | | $R^2 = 0.056$ | |
| | F = 19.028 *** | | F = 7.793 *** | | F = 13.892 *** | | F = 8.024 *** | | F = 17.837 *** | | F = 3.123 ** | |
| **Model 2** | | | | | | | | | | | | |
| Age | −0.045 | −0.129 ** | −0.045 | −0.124 ** | −0.030 | −0.103 * | −0.038 | −0.115 ** | −0.027 | −0.133 ** | −0.019 | −0.060 |
| Sex [1] | 0.386 | 0.042 | 1.691 | 0.173 *** | −0.179 | −0.023 | 0.606 | 0.068 | 0.330 | 0.061 | 0.140 | 0.016 |
| COVID-19 patients [2] | −0.069 | −0.007 | −0.084 | −0.008 | −1.073 | −0.139 ** | −0.401 | −0.044 | 0.006 | 0.001 | −0.668 | −0.077 |
| Suggestions [3] | −1.094 | −0.118 ** | −0.524 | −0.054 | -0.109 | −0.014 | 0.360 | 0.040 | −0.416 | −0.077 | 0.109 | 0.013 |
| Unexpected-events [4] | 1.606 | 0.173 *** | −1.015 | −0.104 * | −1.819 | −0.239 *** | −0.780 | −0.087 * | −0.644 | −0.120 ** | −1.118 | −0.130 ** |
| Instructions [5] | 1.476 | 0.144 ** | 0.714 | 0.066 | 1.345 | 0.160 ** | 1.350 | 0.137 ** | 0.417 | 0.070 | 0.617 | 0.065 |
| PPE [6] | 1.985 | 0.171 *** | 0.767 | 0.063 | −0.143 | −0.015 | 0.403 | 0.036 | 0.647 | 0.096 * | 0.066 | 0.006 |
| Ineffective-decision [7] | −1.101 | −0.070 * | −0.576 | −0.045 | −1.460 | −0.144 ** | −1.613 | −0.136 ** | −1.289 | −0.181 *** | −0.713 | −0.062 |
| Different-behavior [8] | −0.873 | −0.093 * | −0.010 | −0.001 | −0.524 | 0.068 | −0.204 | −0.023 | −1.041 | −0.191 *** | −0.142 | −0.016 |
| DRS | −0.141 | −0.115 ** | −0.304 | −0.235 *** | 0.032 | 0.032 | −0.317 | −0.269 *** | −0.129 | −0.182 *** | −0.117 | −0.103 * |
| | $R^2 = 0.276$ | | $R^2 = 0.181$ | | $R^2 = 0.209$ | | $R^2 = 0.196$ | | $R^2 = 0.282$ | | $R^2 = 0.065$ | |
| | $\Delta R^2 = 0.012$ *** | | $\Delta R^2 = 0.050$ *** | | $\Delta R^2 = 0.001$ | | $\Delta R^2 = 0.065$ *** | | $\Delta R^2 = 0.030$ *** | | $\Delta R^2 = 0.010$ * | |
| | F = 18.156 *** | | F = 10.486 *** | | F = 12.547 *** | | F = 11.622 *** | | F = 18.652 *** | | F = 3.318 *** | |

* $p < 0.05$, ** $p < 0.01$, *** $p < 0.001$; [1] Sex (1 = male; 2 = female); [2] COVID−19 patients (1 = yes; 2 = no); [3] Suggestions (1 = yes; 2 = no); [4] Unexpected-events (1 = yes; 2 = no); [5] Instructions (1 = yes; 2 = no); [6] PPE (1 = yes; 2 = no); [7] Ineffective-decision (1 = yes; 2 = no); [8] Different-behavior (1 = yes; 2 = no); PPE, personal protective equipment, DRS = Dispositional Resilience Scale.

Considering the Pearson's correlations shown in Table 1, to verify Hypothesis 5 hierarchical regression models were used to test the predictive effects of the emergency stress scales and DRS on burnout components, checking by age, sex, and COVID-19 patients for the effects shown by MANOVA analysis (Table 3).

The results showed that hardiness had a negative predictive effect on emotional exhaustion ($\beta = -0.277$, $p < 0.001$) and depersonalization ($\beta = -0.215$, $p < 0.001$), but was a positive predictor of personal accomplishment ($\beta = 0.332$, $p < 0.001$). In step 2, hardiness maintained its predictive effects on burnout, even when emergency stress scales were inserted into the model (Table 3). The results confirmed the effects of hardiness in reducing stress and burnout and increasing personal accomplishment (H5).

**Table 3.** Hierarchical linear regression models on BMI–HSS ($n = 494$).

| | Emotional Exhaustion | | Depersonalization | | Personal Accomplishment | |
|---|---|---|---|---|---|---|
| **Model 1** | **B** | **Exp(B)** | **B** | **Exp(B)** | **B** | **Exp(B)** |
| Age | −0.022 | −0.038 | −0.023 | −0.078 | 0.078 | 0.151 ** |
| Sex [1] | 0.661 | 0.043 | −0.634 | −0.082 | 0.965 | 0.070 |
| COVID-19 patients [2] | −0.342 | −0.022 | −0.125 | −0.016 | −2.352 | −0.168 *** |
| | $R^2 = 0.004$ | | $R^2 = 0.012$ | | $R^2 = 0.043$ | |
| | F = 0.689 | | F = 2.033 | | F = 7.267 *** | |
| **Model 2** | | | | | | |
| Age | 0.006 | 0.010 | −0.012 | −0.040 | 0.048 | 0.093 * |
| Sex [1] | 0.182 | 0.012 | −0.823 | −0.106 * | 1.488 | 0.107 * |
| COVID-19 patients [2] | −0.802 | −0.052 | −0.306 | −0.039 | −1.850 | −0.132 ** |
| DRS | −0.557 | −0.277 *** | −0.221 | −0.215 *** | 0.609 | 0.332 *** |
| | $R^2 = 0.077$ | | $R^2 = 0.056$ | | $R^2 = 0.147$ | |
| | $\Delta R^2 = 0.072$ *** | | $\Delta R^2 = 0.044$ *** | | $\Delta R^2 = 0.104$ *** | |
| | F = 10.090*** | | F = 7.242 *** | | F = 20.957 *** | |
| **Model 3** | | | | | | |
| Age | 0.048 | 0.084 * | −0.002 | −0.007 | 0.037 | 0.071 |
| Sex [1] | −1.134 | −0.074 * | −1.024 | −0.132 ** | 1.690 | 0.122 ** |
| COVID-19 patients [2] | −0.037 | −0.002 | −0.131 | −0.017 | −1.803 | −0.128 ** |
| DRS | −0.197 | −0.087 * | −0.106 | −0.104 * | 0.418 | 0.228 *** |
| Organizational-relational stress | 0.137 | 0.084 | 0.095 | 0.114 * | −0.020 | −0.014 |
| Physical stress | 0.575 | 0.368 *** | 0.048 | 0.060 | 0.084 | 0.059 |
| Inefficacy-decisional stress | −0.029 | −0.015 | −0.044 | −0.044 | 0.301 | 0.166 ** |
| Emotional stress | 0.346 | 0.203 *** | 0.042 | 0.049 | −0.338 | −0.218 *** |
| Cognitive stress | 0.447 | 0.158 ** | 0.307 | 0.213 *** | −0.534 | −0.207 *** |
| COVID-19 stress | 0.078 | 0.044 | 0.009 | 0.010 | 0.002 | 0.001 |
| | $R^2 = 0.491$ | | $R^2 = 0.166$ | | $R^2 = 0.232$ | |
| | $\Delta R^2 = 0.414$ | | $\Delta R^2 = 0.110$ | | $\Delta R^2 = 0.085$ | |
| | *** | | *** | | *** | |
| | F = 46.378 *** | | F = 9.608 *** | | F = 14.506 *** | |

* $p < 0.05$, ** $p < 0.01$, *** $p < 0.001$; [1] Sex (1 = male; 2 = female); [2] COVID-19 patients (1 = yes; 2 = no).

## 4. Discussion

In this study, we aimed to detect the stress perceived by the Red Cross volunteers in the second phase of the Italian pandemic. The main objectives were to verify both the predictive incidence of some stress factors and the protective effect of hardiness in reducing stress levels and burnout. The risk of participants developing burnout in relation to the perceived stress at that stage was also assessed.

Stress levels were found to have high positive correlations with the emotional exhaustion and depersonalization components of burnout, and a significant negative correlation with personal accomplishment in the burnout inventory and hardiness (Table 1). The MANOVA analysis results in this study excluded an effect linked to weekly work hours on stress and burnout levels, which seemed to confirm that the stress reactions of the volunteers are not so much related to the number of shifts

performed but to how the events are perceived. Conversely, according to the results of Morgantini et al. [30], the Red Cross volunteers intervene solely for personal reasons, and they have high motivation in their interventions. They also have more freedom to choose how many hours per week to devote to the service, unlike rescue workers.

The participants in this study were Red Cross volunteers, and, based on their specialization and training as emergency and rescue workers, they were assigned during the COVID-19 emergency to various services or intervention projects: health, social, and emergency. The results of the MANOVA analysis described no effects related to the type of project or service performed, whereas the variables that were found to have significant effects within the sample were those connected to having directly treated COVID-19 patients, age, and sex. Having performed interventions on COVID-19 led to higher levels of stress among volunteer emergency workers, especially in terms of a sense of ineffectiveness in decision-making processes related to choices, evaluations, or methods of intervention (Table 2) [9,10,66–68]. Having offered their help and support to COVID-19 patients seems to have led to a greater sense of personal accomplishment (Table 3). Participants, precisely because they are volunteers, provide their service out of personal motivation and are likely to feel a sense of gratification when they perceive that their intervention has helped others, even if this often pushes them toward a greater risk of emotional exhaustion and burnout, as occurs in the helping professions [38,69].

As recognized by the literature, women tend to develop greater emotional responses to stressful events and are at greater risk of experiencing emotional exhaustion [31,33]. MANOVA and regression analyses showed a significative effect of the female sex on physical and emotional stress and emotional exhaustion (Tables 2 and 3). Women also reported greater personal accomplishment for their interventions during the pandemic. As noted by other studies [8,9,53,70], women involved in rescue interventions during the COVID-19 pandemic showed greater levels of fatigue, sleep disturbances, and physical ailments than men. Because of a greater empathic component that generally distinguishes women, they tend to show a greater sense of personal accomplishment in perceiving themselves as effective in providing adequate care and help to others [71].

From MANOVA analysis, older volunteer workers appeared to show higher stress levels than younger workers, which is in line with the findings in other studies [9,72]. Significant differences emerged with respect to age in terms of organizational, physical, and emotional stress. Older volunteer workers probably perceive a greater sense of responsibility at organizational–relational and emotional levels, both with respect to the service to be guaranteed to the community and with respect to younger volunteer colleagues belonging to the service. Age can also be associated with a greater awareness of the risks, leading to a higher level of emotional and psychophysical tension. The sense of awareness and personal experience that increase with age allow a better knowledge of one's own resilience skills other than hardiness. Hardiness and high personal accomplishment seem to characterize the experience of younger volunteers, who tend to experience more enthusiasm and a sense of initiative and challenge or opportunity toward new events, favoring a greater sense of hardiness and personal accomplishment. The association between hardiness and personal accomplishment found in this study confirms the findings reported by Bartone et al. [52].

Volunteer workers who directly treated COVID-19 patients experienced higher stress levels, especially major inefficacy-decisional stress; however, they perceived more personal accomplishment because they performed their intervention specifically on COVID-19 patients. According to several studies [21,22], high stress levels and a sense of personal exhaustion lead to perceiving one's own energy resources as lower, to reducing personal accomplishment, and to losing positive feelings about oneself and others [23].

Risk factors considered in the present study increased the emergency stress levels, supporting Hypothesis 2. The results reported in Table 2 show the significant predictivity of a lack of PPE, and its effect on stress has already been highlighted by other studies [9,30,55], and of experiencing unexpected events. Even though the data were collected in phase 2 of COVID-19, which is when specific intervention protocols had already been implemented and when knowledge of the infection

had been created, the participants reported difficulties affecting the organization due to the lack of PPE. Notably, in phase 2 of the pandemic, sufficient PPE for all Italian operators was lacking, including for emergency operators. Higher organizational stress leads to negative feelings and burnout [34].

As often happens in emergency situations, rescuers find themselves having to manage unexpected events, which leads them to increased use of their internal resources, causing higher stress reactions, thereby augmenting the risk of developing the emotional exhaustion and depersonalization of burnout. Unexpected-events showed a significative predictive effect on the all emergency stress scales (Table 2) and higher stress levels increased the risk of developing burnout (Table 3).

The results showed that the lack of adequate and necessary instructions to be able to intervene in a timely manner seems to have had a significant impact on the levels of stress, especially at the organizational–relational and emotional levels, and on the sense of effectiveness in making decisions (Table 2) [8].

Unlike the study by Maiorano et al. [73] conducted during the first phase of the lockdown, in the present study, the stress factors related to counterfactual thinking seem to have had a negative impact, leading to an increase in stress levels. During phase 2 of COVID-19, i.e., when emergency workers had intervention protocols and greater knowledge of the phenomenon available and the numbers of infected people and deaths were significantly lower, emergency workers were probably able to have more time to reflect on their operational choices and the methods of their interventions, leading them to think in terms of hindsight. This type of thinking, as pointed out by many authors, can be associated with feelings of guilt, frustration, regret, and a sense of ineffectiveness for how one has acted previously [11,12]. These results may indicate a capacity for self-criticism, debriefing, and monitoring of their intervention in the emergency workers; however, considering their association with higher levels of stress, the results seem to indicate the need to provide support and interventions to prevent possible reactions of emotional exhaustion and reduction of one's personal accomplishment. These feelings and reactions, however, can be associated with a lower sense of self-efficacy, also reducing the use of one's own resources and resilience capacity. In this sense, the factors considered in this study, such as "having made a decision that proved ineffective," "having suggested solutions that have not been considered," and "in hindsight, believing that it would have been appropriate to intervene in a different way," are risk factors that significantly impact stress levels and, in particular, the inefficacy-decisional stress scale of emergency workers (Table 2).

Hardiness appears to play a protective role with respect to both stress levels and the risk of developing burnout, demonstrating the negative predictive effects, thereby supporting Hypotheses 4 and 5 (Tables 2 and 3). Hardiness, as noted by other studies [46,48], allows people to promote active attitudes, to be committed to achieving a goal, and to perceive external situations, even negative ones, as an opportunity for challenge. As highlighted by other studies conducted on healthcare and emergency workers involved during the COVID-19 pandemic, hardiness prevents the development of stress, vicarious trauma, and other psychological symptoms [8,9,50,73–75], as well as burnout, as shown in Table 3. The results shown in Table 3 highlight how hardiness demonstrated a significant effect in reducing the levels of emotional exhaustion and depersonalization and simultaneously reinforcing the sense of personal accomplishment, which appears important, especially in helping professions to counteract the risk of developing burnout. In particular, the model presented in Table 3 concerning personal accomplishment emerged as the main effect, compared to the other variables, explaining 10% of the variance (see step 2 of Table 3). Confirming Hypothesis 5, Table 3 shows how hardiness had a negative predictive effect on the components of burnout, explaining 7% and 4% for emotional exhaustion and depersonalization, respectively. Hardiness maintained its effects on all burnout components, even when the emergency stress scales were included in the analysis models.

The presence of hardiness in the model reduced the effects of the other variables such as age and having treated COVID-19 patients. According to several studies, hardiness may mitigate and reduce the effect of stress and burnout on mental health [40–44,51] and can increase the feeling of personal accomplishment [38], which seems to be higher in younger than in older emergency workers.

The results of this study are in line with research conducted on previous pandemics with characteristics similar to COVID-19, such as SARS and MERS, which reported high levels of stress and burnout in health and emergency workers employed in the management of health emergencies [29,76–79]. Studies that focused on health workers employed in emergency departments during the MERS epidemic have highlighted how stress levels were positively associated with burnout and that burnout level was significantly higher in emergency department nurses who cared for MERS-CoV-infected or MERS-CoV-suspected patients. In line with our findings, other risk factors associated with a high level of stress and burnout during both the SARS [29] and MERS [77,78] epidemics were female sex, age, as well as organizational factors such as the availability of PPE [78]. Finally, studies conducted on nurses during the SARS epidemic highlighted the role of hardiness as a protective factor on the mental health of nurses [79].

These results indicate how, with a view to prevention, providing forms of support to rescuers during the emergency phases is advisable to restore their resilience and their ability to cope with the risk factors.

The World Health Organization and International Labour Organization (WHO and ILO, 2018) [80] underlined that, during emergency management like that of a pandemic, close coordination and cooperation among a large number of different organizations is required for workers and responders (healthcare and professional groups, including firefighters, police officers, personnel, emergency doctors, and humanitarian aid). These organizations stated that those working in these emergency situations are at risk of stress and developing burnout. The results obtained in this study confirmed this risk and the importance of guidelines for managing the response to work-related stress of the people involved through developing policies in several areas: pre- and post-deployment screening, assessment of the capacities of staff to respond to stressors, appropriate training in managing stress, and support interventions.

The study sample can be considered representative of the Italian Veneto region and, considering what was highlighted by the World Health Organization, the results obtained allow for a sufficient generalization, although further confirmation through additional studies is needed.

At present, no similar studies have yet been conducted, which represents the originality of the results reported above. However, the absence of other studies may limit the level of generalization of the results.

## 5. Limitations

There are some limitations to this study. The first limitation is that the research was conducted through an online platform, so the compilation of the questionnaires was not structured and was probably influenced by distraction factors. A second limitation is that this was a cross-sectional study, whereas a longitudinal study would allow for a better analysis of phenomena such as the development of symptoms of burnout. A third limitation is the use of a self-reported questionnaire and the participants' lack of knowledge about the presence of previous psychological problems. Finally, the study lacks a comparison with other emergency workers involved throughout the national territory and belonging to other organizations, limiting the generalization of the results.

## 6. Conclusions

We aimed to illustrate the difficult situation faced by volunteer emergency workers during the second phase of the COVID-19 pandemic in Italy. The results of this study highlighted how emergency workers who worked with COVID-19 patients experienced high stress levels. It is necessary to implement immediate interventions that increase the activation of protective factors that can mitigate and prevent the development of serious psychological consequences and burnout. Immediate interventions are essential to activate psychological resilience and guide workers in the use of the most effective long-term coping strategies to protect their mental health. Doing so may reinforce, for example, a greater sense of effectiveness and personal accomplishment, and reduce

aspects such as emotional exhaustion and depersonalization, which are associated with burnout situations. Furthermore, prevention interventions must be developed to guide volunteers to better deal with organizational difficulties, interpersonal tensions, and a sense of helplessness. To this end, it will be useful to develop hardiness and resilience training for volunteer emergency workers [81–83]. Hardiness training should include providing information on hardiness and analyzing case studies with an emphasis on detecting threats, coping strategies, and stress management concepts [84]. Providing this type of intervention is crucial for the effective management of public health epidemics like COVID-19, which should be a priority for health organizations.

The preparation and effectiveness of health care organizations to manage these crises over indefinite periods of time can produce considerable differences in the impact of the pandemic on citizens' well-being. Thus, by learning new operating strategies, volunteer emergency workers might mitigate their burnout, and by making organizational supports congruent with emergency workers' specific needs, healthcare organizations will optimize their health crisis management and have the extra advantage of improving the lives of their workers, and therefore the level of care they provide to the recipients of their services.

**Author Contributions:** Conceptualization, M.V., T.M., V.G., and D.P.; methodology, M.V., T.M., V.G., and G.S.; validation M.V., T.M.; formal analysis M.V., T.M.; data curation, M.V., T.M.; writing—original draft preparation M.V., V.G., T.M., and G.S.; writing—review and editing, M.V., V.G., and T.M.; visualization, V.M. and T.M.; project administration, M.V., D.P. All authors have read and agreed to the published version of the manuscript.

**Funding:** This research received no external funding.

**Conflicts of Interest:** The authors declare no conflict of interest.

## Appendix A

*Additional Analysis*

Table A1 shows the means and standard deviations from the ESQ, DRS-15, and MBI–HSS. Table A2 shows distributions of risk factors.

**Table A1.** Average scores on ESQ, DRS, and MBI–HSS ($n$ = 494).

|  | Mean (SD, Min–Max) |
| --- | --- |
| **ESQ** | |
| Organizational-Relational Stress | 11.76 (4.62, 4–30) |
| Physical Stress | 6.02 (4.82, 0–20) |
| Inefficacy-decisional Stress | 8.91 (3.78, 0–20) |
| Emotional Stress | 7.94 (4.43, 0–20) |
| Cognitive Stress | 4.48 (2.66, 0–14) |
| COVID-19 Stress | 10.14 (4.26, 0–20) |
| **Total DRS** | 28.09 (3.75, 14−37) |
| **MBI–HSS** | |
| Emotional-Exhaustion | 8.06 (7.54, 0–40) |
| Depersonalization | 3.18 (3.87, 0–20) |
| Personal-Accomplishment | 28.76 (6.89, 0–42) |

ESQ, Emergency Stress Questionnaire; DRS, Dispositional Resilience Scale; MBI–HSS, Maslach Burnout Inventory–Human Services Survey.

**Table A2.** Risk factor distributions in volunteer emergency workers ($n = 494$).

| Risk factor | Present ($n$) | Absent ($n$) | $\chi^2$ |
|---|---|---|---|
| Suggestions | 217 | 277 | 7.287 ** |
| Unexpected-events | 275 | 217 | 6.837 ** |
| Instructions | 356 | 138 | 96.202 *** |
| PPE [5] | 399 | 94 | 188.692 *** |
| Ineffective-Decision | 82 | 410 | 218.667 *** |
| Different-behavior | 297 | 197 | 270.198 *** |

** $p < 0.01$; *** $p < 0.001$.

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
