# Peer review of "Personal Accomplishment and Hardiness in Reducing Emergency Stress and Burnout among COVID-19 Emergency Workers"

_sustainability, doi:10.3390/su12219071_

Round 1

Reviewer 1 Report

Thank you for the opportunity to review the paper “Personal Accomplishment and Hardiness in 3 Reducing Emergency Stress and Burnout among 4 Covid-19 Emergency Workers”. It is an interesting topic and worth of wide discussion for reducing emergency stress and burnout in the context of the COVID-19 pandemic. However, there are still some concerns of this paper. Following are my comments of this paper. Hope the comments may be helpful to the author(s).

In Abstract the authors should avoid using abbreviations (e.g. PPE).

Line 100: the authors should explain this abbreviation: HCP.

Methodology and results:

Hypotheses must be formulated in close relation with the literature, eventually in the same section (e.g. in the Literature review).

Hypothesis H1 should be reformulated and divided in 2 hypotheses since it refers to 2 different analyses/ tests (positive correlations/ negative correlations). The same situation is for hypothesis H2.

I didn’t understand the statement „In the Hypothesis 2 and 3 other three variables, gender, age and Covid-19 patients were included.” How the three variables were included? What is their role in the analysis? Are they a part of those hypotheses?

In section Results, the authors need to explain in more detail/ more clearly the steps of the analysis, the significance of the results, emphasizing those findings which support/ reject the hypotheses formulated previously. In addition, what are the results which confirm/ reject the three hypotheses? In addition, I would like to see an interpretation of the (most important) figures in the tables.

I suggest the authors to correlate more closely the results obtained with the information in the section Discussion.  The authors could explicate some findings  in the section Discussion by mentioning the results from the previous section which support those statements. For example, the authors mentioned that „Stress levels have a high positive association with the emotional exhaustion and depersonalization component of burnout, while they have a significant negative correlation with the personal accomplishment of burnout inventory and hardiness (see Table 1).” What are the results/ figures that indicate the positive association/ the significant negative correlation? The authors should make an effort to replace the reference to tables („see Table X”) with figures/ results and their significance that explain and support the argument.

There are some typos in the manuscript, please proofread.

Author Response

Thank you for the opportunity to review the paper “Personal Accomplishment and Hardiness in 3 Reducing Emergency Stress and Burnout among 4 Covid-19 Emergency Workers”. It is an interesting topic and worth of wide discussion for reducing emergency stress and burnout in the context of the COVID-19 pandemic. However, there are still some concerns of this paper. Following are my comments of this paper. Hope the comments may be helpful to the author(s).

REPLY: The authors thank the reviewer for the helpful suggestions provided.

In Abstract the authors should avoid using abbreviations (e.g. PPE).

REPLY: The explanation of PPE has been added.

Line 100: the authors should explain this abbreviation: HCP.

REPLY: The explanation of HCP has been added.

Methodology and results:

Hypotheses must be formulated in close relation with the literature, eventually in the same section (e.g. in the Literature review).

Hypothesis H1 should be reformulated and divided in 2 hypotheses since it refers to 2 different analyses/ tests (positive correlations/ negative correlations). The same situation is for hypothesis H2.

REPLY: The hypothesis section has been re-organized into five points.

I didn’t understand the statement „In the Hypothesis 2 and 3 other three variables, gender, age and Covid-19 patients were included.” How the three variables were included? What is their role in the analysis? Are they a part of those hypotheses?

REPLY: The sentence has been modified and the role of age, gender and COVID-19 patients variables has been specified.

In section Results, the authors need to explain in more detail/ more clearly the steps of the analysis, the significance of the results, emphasizing those findings which support/ reject the hypotheses formulated previously. In addition, what are the results which confirm/ reject the three hypotheses? In addition, I would like to see an interpretation of the (most important) figures in the tables.

I suggest the authors to correlate more closely the results obtained with the information in the section Discussion.  The authors could explicate some findings in the section Discussion by mentioning the results from the previous section which support those statements. For example, the authors mentioned that „Stress levels have a high positive association with the emotional exhaustion and depersonalization component of burnout, while they have a significant negative correlation with the personal accomplishment of burnout inventory and hardiness (see Table 1).” What are the results/ figures that indicate the positive association/ the significant negative correlation? The authors should make an effort to replace the reference to tables („see Table X”) with figures/ results and their significance that explain and support the argument.

REPLY: More information on the analysis conducted, significance of the results and on the data confirming the hypotheses has been added in the results and discussion sections.

There are some typos in the manuscript, please proofread.

REPLY:  The paper had been revised for editing and English by the English Editing company, which was asked for further control and revision after this comment.

Reviewer 2 Report

The manuscript “Personal Accomplishment and Hardiness in  Reducing Emergency Stress and Burnout among Covid-19 Emergency Workers” uses primary data obtained from questionnaires from 494 emergency volunteers from the Red Cross Committee in Veneto, engaged in various health, emergency, and social activities aimed at Covid-19 patients and people at risk of contracting the virus.

The study analyses the effects of risk and protective factors on burnout. They also analyse the influence of age, gender, weekly hours of service, stress risk factors, and use of PPE.

In general, arguments are clearly exposed. Literature review is appropriate. Hypotheses and methods used are well presented. Please, clarify if the sample of respondents properly represents the distribution of emergency workers in Veneto.

Results, conclusions and limitations are adequately explained. Although the paper would improve with an extended comparison of results found during the covid-19 pandemic compared to other emergency situations. 

Also, figures in pages 6-7 with results from tests (lines 254-276) would be better in a table.

More policy implications could be added. 

Revise the spelling and verb tenses. For instance, some mistakes on lines 12, 68, 115…

Author Response

The manuscript “Personal Accomplishment and Hardiness in  Reducing Emergency Stress and Burnout among Covid-19 Emergency Workers” uses primary data obtained from questionnaires from 494 emergency volunteers from the Red Cross Committee in Veneto, engaged in various health, emergency, and social activities aimed at Covid-19 patients and people at risk of contracting the virus.

The study analyses the effects of risk and protective factors on burnout. They also analyse the influence of age, gender, weekly hours of service, stress risk factors, and use of PPE.

In general, arguments are clearly exposed. Literature review is appropriate. Hypotheses and methods used are well presented. Please, clarify if the sample of respondents properly represents the distribution of emergency workers in Veneto.

Results, conclusions and limitations are adequately explained. Although the paper would improve with an extended comparison of results found during the covid-19 pandemic compared to other emergency situations. 

REPLY: The authors thank the reviewer for the helpful suggestions provided

Also, figures in pages 6-7 with results from tests (lines 254-276) would be better in a table.

Thanks to the reviewer for the suggestion. However, the authors believe that these data are better explained in the text given that they are results of MANOVA analysis and explaining the results both in a table and in the text would be too redundant and not in line with the APA norms.

More policy implications could be added. 

REPLY: Policy implications were added in the conclusion section.

Revise the spelling and verb tenses. For instance, some mistakes on lines 12, 68, 115…

REPLY:  The paper had been revised for editing and English by the English Editing company, which was asked for further control and revision after this comment.

Reviewer 3 Report

The paper refers to an important research problem (burnout among Covid-19 emergency workers) and it is well-positioned in Journal scope. From a methodical and practical point of view, the investigated problem is interesting.

The structure of the document is transparent. Methodical contributions (survey research, statistical analysis) and the results have great practical significance. In my opinion, the methodological apparatus used is adequate to the problem being addressed.

Importantly, the authors also recognize the limitations of the study. Questionnaire form structure, target respondent group, form of proceeding the survey, the time of the survey, place, they influence the answers obtained. It is important to continue and improve this research, because the problem is considered important.

Whether the results of these studies can be generalized (other groups of medical workers, emergency services, fire services, police, etc.)?
If so, how?

Author Response

The paper refers to an important research problem (burnout among Covid-19 emergency workers) and it is well-positioned in Journal scope. From a methodical and practical point of view, the investigated problem is interesting.

The structure of the document is transparent. Methodical contributions (survey research, statistical analysis) and the results have great practical significance. In my opinion, the methodological apparatus used is adequate to the problem being addressed.

Importantly, the authors also recognize the limitations of the study. Questionnaire form structure, target respondent group, form of proceeding the survey, the time of the survey, place, they influence the answers obtained. It is important to continue and improve this research, because the problem is considered important.

REPLY: We thank the reviewer for the appreciation of our paper and for the interesting comments.

Whether the results of these studies can be generalized (other groups of medical workers, emergency services, fire services, police, etc.)?
If so, how?

REPLY: We added some information on the generalization of the results but, at the same time, the impossibility of a complete generalization in the limits section was highlighted.

Round 2

Reviewer 1 Report

I have no other comments related to the work.

I wish the authors all the best.

This manuscript is a resubmission of an earlier submission. The following is a list of the peer review reports and author responses from that submission.